# Reimplementation of FixMatch and Investigation on Noisy (Pseudo) Labels and Confirmation Errors of FixMatch

## Reproducibility Summary

**Scope of Reproducibility.**    The main objective of this work is to confirm the effectiveness of FixMatch (Sohn et al. [2020]), which combines pseudo labelling and consistency regularization in semi-supervised learning (SSL) tasks, by achieving similar results on CIFAR-10 and demonstrating the key success of FixMatch via ablation studies. Furthermore, we also investigated the the existence of confirmation errors in FixMatch by reconstructing the batch structure during the training process.

**Methodology.**    All the experiments in this work were conducted on CIFAR-10 using the same network architecture, Wide ResNet28-2. A single V100 is used for each experiment with an average training time of 70 hours. We re-implemented FixMatch mainly based on the paper using Pytorch and refer to the official implementation (in Tensorflow) for details and replicated similar results shown in the second-last row of Table 2 of column CIFAR-10 in Sohn et al. [2020]. Ablation studies were focused on two key factors of FixMatch, ratio of unlabeled data and confidence threshold, as shown in Figure 3 (a) & (b) in Sohn et al. [2020].

**Results.**    Compared with the average error rate reported in Table 2 in Sohn et al. [2020], our implementation achieves similar error rates by 3.77 lower on CIFAR-10 with 40 labels, 0.22 higher on CIFAR-10 with 250 labels, and 0.1 higher on CIFAR-10 with 4000 labels. Thus it is supported that FixMatch outperforms semi-superivesed learning benchmarks. And the results of ablation studies exhibit almost the same trends as Figure 3 (a) & (b) show in the paper, which demonstrated that the author's choices with respect to those ablations were experimentally sound. We also confirmed the existence of confirmation errors in pseudo labels by checking the confusion matrix of the prediction of unlabeled data in different training stages.

**What was easy.**    It is generally easy to re-implement FixMatch given all the experimental settings in the paper, with key parameters clearly stated in each experimental section and detailed lists of hyperparameters in appendix. Compared with CTAugment, RandAugment is relatively easy to implement since it requires no parameters tuning during training and coefficients representing the severity of all distortions are given in appendix. Besides, it converges faster than CTAugment.

**What was difficult.**    The official implementation is complicated thus not easy to follow. And there are some details missing in the paper compared to the code: 1. the official implementation actually use leaky ReLU instead of ReLU for ResNet; 2. Exponential moving average is only mentioned for experiments on ImageNet but actually also used on CIFAR-10; 3. the details on how to update the weights of the magnitude bins of CTAugment are not given in the paper, and our implementation achieves a slightly worse results than the average error rate reported (1.14 higher with 250 labels).

**Communication with original authors.**    All the confused parts mentioned in the previous section are clarified by the original authors via email and in the issues of their Github repository.

Submitted to ML Reproducibility Challenge 2020. Do not distribute.

# Abstract

FixMatch is a semi-supervised learning method, which achieves comparable results with fully supervised learning by leveraging a limited number of labeled data (pseudo labelling technique) and taking a good use of the unlabeled data (consistency regularization ). In this work, we reimplement FixMatch and achieve reasonably comparable performance with the official implementation, which supports that FixMatch outperforms semi-superivesed learning benchmarks and demonstrates that the author's choices with respect to those ablations were experimentally sound. Next, we investigate the existence of a major problem of FixMatch, *confirmation errors*, by reconstructing the batch structure during the training process. It reveals existing confirmation errors, especially the ones caused by *asymmetric noise* in pseudo labels. To deal with the problem, we apply equal-frequency and confidence entropy regularization to the labeled data and add them in the loss function. Our experimental results on CIFAR-10 show that using either of the entropy regularization in the loss function can reduce the asymmetric noise in pseudo labels and improve the performance of FixMatch in the presence of (pseudo) labels containing (asymmetric) noise. Our code is available at the url: https://github.com/Celiali/FixMatch.

# 1 Introduction

Ghahramani [2020] summarized the reasons for the success of deep learning in his talk given as the chief scientist in Uber. Firstly, with the availability of large datasets, large models can work well. Secondly, training such large models with stochastic descent works surprisingly well. Moreover, staying close to identity (such as ReLU) makes it stable to be trained. The automate differentiation and a large number of open source softwares make it scale well. Therefore, we can see deep learning in many applications, such as computer vision, natural language processing, bioinformatics, etc.

However, it is not always the case where a huge number of labeled data are available. In some areas, it is difficult, expensive, or even impossible to have a large labeled dataset, such as medical images [Kuznetsova et al., 2018]. In this case, it can be difficult to train a Deep Neural Network (DNN) from scratch with the limited labeled data. Luckily, Tajbakhsh et al. [2016] shows that a DNN trained based on a pre-trained DNN, fine-tuning, can outperform the one trained from scratch. Moreover, Semi-Supervised Learning (SSL) is also a common method to deal with the scarcity and often high acquisition cost of labelled data [von Kügelgen et al., 2020]. SSL efficiently leverages labeled data and the relation with unlabeled data to train a DNN. Among SSL methods, there is a class of "match"-based methods, such as FixMatch [Sohn et al., 2020], MixMatch [Berthelot et al., 2019], ReMixMatch [Kurakin et al., 2020] and DivideMatch [Li et al., 2020]. These methods utilize the consistency regularization, pseudo-labelling and ensembling methods to boost the performance with the use of unlabeled data. In fact, they are leveraging prior knowledge to regularize the training of DNNs. In this project, we focus on reproducing and investigating one of such methods, FixMatch [Sohn et al., 2020].

Nevertheless, SSL is still facing many challenges in theory and in practice. Ben-David et al. show that "as long as one does not make any assumptions about the behavior of the labels, SSL cannot help much over algorithms that ignore the unlabeled data." Moreover, SSL can actually degrade performance if certain assumptions are not met [Chapelle et al., 2010]. In this line of works, Schölkopf et al. [2012] consider the problem from a causal modeling perspective and conclude that in fact SSL is impossible when predicting a target variable from its causes (causal learning) but possible from anti-causal learning. Recently, the relation of causality and semi-supervised learning is further explored in [von Kügelgen et al., 2020], i.e., predicting a target variable from both causes and effects at the same time. Moreover, in the light of consistency regularization and pseudo-labelling, a significant issue of the "Match"-based methods is *confirmation error*. It happens especially when noisy samples are in the labeled set. A DNN can keep having lower loss by fitting the noise and be further maintained after training with the wrong pseudo labels of unlabeled data , which keeps the errors in the model and limits its generalization and performance [Tarvainen and Valpola]. This problem becomes more serious in the presence of asymmetric noise in the training labels, which roughly speaking tends to label a class of data as another specific class.

Therefore, in this work, we are not only reimplementing FixMatch, but also investigating whether the pseudo labels made by the DNN contain harmful noise leading to confirmation errors. First, we design a stable and reliable method to examine the existence of confirmation errors and noisy pseudo labels by reconstructing the batch structure. Secondly, we find methods to deal with (asymmetric) noise in (pseudo) labels of the training dataset. We reconstruct the batch structure and add an equal-frequency entropy regularization on labeled data to the loss function of FixMatch. Moreover, we also use a confidence entropy regularization on labeled data to avoid the over-confident prediction. It turns out that both entropy regularization is helpful for dealing with the noisy (pseudo) labels (even for the asymmetric noise) and confirmation errors. Our experimental results show that

1. our implementation can achieve almost the same performance even better for low-label regimes.

2. there exists asymmetric noise in the pseudo labels leading to confirmation errors. With such pseudo labels, the model is biased which in turn leads to more asymmetric noise in pseudo labels.

3. FixMatch with equal-frequency entropy regularization and FixMatch with confidence entropy regularization can reduce (asymmetric) noise in the pseudo labels and perform better than the baseline FixMatch in the presence of asymmetric noise in (pseudo) labels .

## 2 Related work

As introduced in Sec. 1, confirmation error is a serious issue of "Match"-based SSL methods and our study is mainly about the confirmation error and FixMatch in the presence of noisy (pseudo) labels. Therefore, here we mainly introduce the noisy labeling and some related works for dealing with the noisy label and confirmation error in SSL.

**Noisy labeling and noise-robust loss.** Suppose a dataset $\mathcal{D} = \{(x_i, y_i)\}_{i=1}^{n}$ where $y_i$ is given by noisy labeling. To model noisy labeling process, we have $p(y_i|\widetilde{y_i})$ where $\widetilde{y_i}$ is the ground truth label under the assumption that the noise label is conditionally independent from the input data given the ground-true label; formally, $p(y_i = k|x_i, \widetilde{y_i} = j) = p(y_i = k|\widetilde{y_i} = j) = \eta_{kj}$. In general, such noise is called class dependent, which is also named as the asymmetric noise[Zhang and Sabuncu, 2018]. In contrary, when $\eta_{kj} = \eta$, it is called symmetric noise. Under the symmetric noise assumption, Ghosh et al. [2015] studied the functional form of loss function and concluded that by using the symmetric loss function, one can get a global optima such that the learned model is noise tolerant. For example, the MAE loss function is a symmetric function while the cross entropy loss function is not. However, using MAE loss function has poor accuracy performance on classification tasks compared with the cross entropy loss function [Zhang and Sabuncu, 2018]. One can convince oneself with Eqn. (5) in [Zhang and Sabuncu, 2018], i.e., the cross entropy loss function enables the optimization process weighting the sample importance while the MAE loss function considers samples equally. Furthermore, Zhang and Sabuncu [2018] combine MAE and cross entropy loss functions with L'Hôpital's rule, i.e.,

$$\mathcal{L}_q(f(x), j) = \frac{(1 - f_j(x)^q)}{q}, \tag{1}$$

where $f(x)$ is the model, $j$ indexes the class, and $f_j(x)$ is the softmax output of $j$. Interestingly, when $q = 1$ , $\mathcal{L}_q(f(x), j)$ is a MAE loss function; while $\lim_{q \to 0} \mathcal{L}_q(f(x), j)$ is a cross entropy loss. Therefore, one can manipulate trade off by selecting a good hyper-parameter $q$. Furthermore, it also introduces a better loss function, the truncated $\mathcal{L}_q(f(x), j)$, which is essentially a practically improved version of $\mathcal{L}_q(f(x), j)$. However, in theory the proposed method is based on the symmetric noise assumption [Zhang and Sabuncu, 2018], which can be quite easy to be violated. This is a trade-off between using a stricter assumption and estimating noisy labelling mechanisms [Patrini et al., 2017] (which is a challenge).

**SSL for noisy labeling and a potential solution for asymmetric noise.** Li et al. [2020] consider the noisy label problem as a semi-supervised learning problem by finding the similarity of unlabeled samples in semi-supervised learning and noisy labels. Suppose that we can successfully separate the noisy and clean samples, we can treat the noisy ones as unlabeled data in semi-supervised learning, and then leverage the success of semi-supervised learning to tackle the noisy labeling problem.

139   Firstly, by observing that the loss of clean samples tends to be lower than the noisy ones [Arazo et al.,
140   2019], Li et al. [2020] fit a Gaussian Mixture Model for the two components, the noisy group and
141   the clean one. Then given a loss, it can be inferred whether the sample is a noisy one or a clean
142   one. Consequently, following the mentioned idea, semi-supervised learning methods are applied
143   to such a separated dataset. Moreover, Li et al. [2020] consider the influence of asymmetric noise
144   in the supervised learning phase. Because the bias introduced by the asymmetric noise can lead
145   to severe consequences (confirmation errors). [Li et al., 2020] added a negative entropy penalty
146   term $-\mathcal{H} = \sum_j f_j(x) \log f_j(x)$ for an input $x$ in the cross-entropy loss function at the beginning
147   of training to avoid over-confident prediction, which works well emperically. To further reduce
148   the influence of the confirmation error introduced by the symmetric noise, it uses the MixMatch
149   [Berthelot et al., 2019] procedure to train two independent DNNs and attractively exchange datasets
150   with each other for filtering errors made by the other one. This is actually an ensemble method, which
151   reduces the random noise in the prediction, especially in the presence of symmetric labelling noise.

152   **Model bias in SSL.**   Kurakin et al. [2020] propose a distribution alignment method utilizing a
153   principle introduced by Bridle et al. [1992]. It formulates an ideal classifier which maximizes the
154   mutual information of model inputs and model outputs. Furthermore, it argues that the second term
155   of mutual information encourages a model to output with low entropy and high confidence, while
156   another one encourages equal frequency across the entire training set as shown in

$$
\begin{aligned}
\mathcal{I}(y; x) &= \iint \log \frac{p(y, x)}{p(y)p(x)} dy dx \\
&= \mathcal{H}[\mathbb{E}[p(y \mid x; \theta)]] - \mathbb{E}_x[\mathcal{H}[p(y \mid x; \theta)]],
\end{aligned} \tag{2}
$$

157   where $\theta$ is the model parameters. As what Kurakin et al. [2020] said, when the marginal distribution
158   of a training dataset labels is not uniformly distributed, it is not proper to regularize the frequency. In
159   our work, to deal with such case, we augment the training dataset and make the labels of labeled data
160   in each batches to be uniformly distributed.

## 3   Methods

### 3.1   FixMatch

163   As one of the SSL methods, FixMatch [Sohn et al., 2020] leverages labeled data and introduces prior
164   knowledge about unlabeled data in the training process. For labeled data, FixMatch simply uses the
165   cross entropy loss function for a batch,

$$
l_s = \frac{1}{B} \sum_{b=1}^{B} H(y_b, f(\alpha(x_b))), \tag{3}
$$

166   where $B$ is the number of labeled data in a batch, $x_b$ is a labeled sample, $y_b$ is the label, and $\alpha(\cdot)$ is
167   weak augmentation. However, due to limited number of labeled samples, the performance of such
168   DNN is not ideal. Therefore, the question is how to make a good use of the sufficient unlabeled data
169   to improve the performance? Ideally, the performance can be close to the DNN trained with the fully
170   labeled dataset.

171   FixMatch considers the consistency of model prediction on the unlabeled data with weak and strong
172   augmentation (the augmentation methods are introduced in Sec. 4). It first uses the model to predict
173   pseudo labels for unlabeled data and then compute the loss of unlabeled data with the pseudo labels
174   and the consistency regularization. The loss function for the unlabeled samples $u_b$ is

$$
l_u = \frac{1}{\mu B} \sum_{b=1}^{\mu B} \mathbf{1}(\max(f(\alpha(u_b))) \geq \tau) H(\hat{y}_b, f(\mathcal{A}(u_b))), \tag{4}
$$

175   where $\mu B$ is the number of unlabeled data in a batch, $\hat{y}_b := \arg\max_y p(y|\alpha(u_b); \theta_f)$ is the pseudo
176   label of $u_b$, $\theta_f$ is the neural network parameters of function $f$, and $\mathcal{A}(\cdot)$ is the strong augmentation.
177   Note that to make pseudo labels reliable to be used, FixMatch considers the pseudo labels in the loss
178   function only if the prediction has a higher probability than $\tau$. Next, together with the cross entropy
179   loss of labeled data, the loss function of FixMatch is $l_s + \lambda_u l_u$.

## 3.2 Investigation of noisy (pseudo) labels and confirmation errors of FixMatch

**Nosiy pseudo labels and confirmation errors in FixMatch.** A main issue of "Match"-based SSL methods is confirmation errors. Since FixMatch is trained on batches with both labeled and unlabeled data, it is very likely to make prediction errors at the beginning of the training. When the model makes wrong predictions of labeled data, since we have their ground-truth labels, the model can become better with the loss for labeled data. But when it comes to unlabeled samples, since we don't have the ground-truth labels, the model uses the confident pseudo labels as the labels for training. In this case, if the pseudo-labels are noisy, the model can fit such errors and become biased. In the next batch, it can generate more wrong pseudo-labels with higher confidence. Moreover, the consistency regularization can keep reinforcing the model to fit such wrongly labeled data. Finally, it demonstrates a biased model with a poor performance on generalization and robustness. Therefore, noise in the pseudo labels can lead to confirmation errors in FixMatch.

Both asymmetric noise and symmetric noise in pseudo labels can lead to confirmation errors, but in general asymmetric noise is more harmful and harder to deal with. For example, to reduce the impact of symmetric noise and get an unbiased model, one can use ensembling methods like [Li et al., 2020] to train multiple DNNs at the same time; however, this can fail in the presence of asymmetric noise. In this work, we focus on asymmetric noise and one can simply extend the method to deal with the influence of symmetric noise with ensemble methods.

**Investigation with class-balanced batches.** To check whether there exist confirmation errors, we need to check that during the training process errors are reinforced by the model. Moreover, to see the asymmetric noise in the pseudo labels, we need to check that in the training phase whether FixMatch predicts a certain class of unlabeled data into certain other classes. Thus, these require us to investigate the performance of FixMatch at each batch and check the pseudo labelling performance regarding asymmetric noise in the pseudo labels. However, in [Sohn et al., 2020], a batch is not necessary to contain all the classes of training dataset and it can contain different classes with different numbers. Therefore, the performance of pseudo labelling regarding asymmetric noise inherits the randomness of batch composition, which makes the investigation conclusion unreliable.

To deal with this issue, we reconstructed the batch structure which requires each batch to contain an equal number of images for all the classes on both labeled and unlabeled data, called Balanced-Class (BC) batches. With such batches, we can fairly check the performance of pseudo labelling in each batch how many errors are made when the model predicts each class and whether it tends to label a class as other certain classes causing asymmetric noise. Note that without further introducing regularization, BC batches on their own cannot improve the performance of FixMatch, which has indistinguishable results without BC as shown in Sec. 5.3.

Furthermore, we leverage the reconstructed batch structure to regularize the training process for reducing the noise in pseudo labels and improving the performance. With the reconstructed batches, we know that the class of labeled data[1] is uniformly distributed, thus we can regularize the output of labeled data with the negative entropy loss of the prediction frequency. In this way we force the output of labeled data to be uniformly distributed. Potentially this can regularize the asymmetric noise in the labeled data because the output class distribution is not likely to be uniformly distributed in the presence of asymmetric noise. Consequently, it can reduce the asymmetric noise in pseudo labels because the prediction on both labeled and unlabeled data uses the same network which is unlikely to have different prediction behavior. Therefore, we add an equal-frequency entropy regularization to the loss function, which is

$$
\begin{aligned}
l' &= l'_s + \lambda_u l_u, \qquad\qquad\qquad\qquad\qquad\qquad\qquad\qquad (5)\\
l'_s &= l_s - \lambda_{ef}\mathcal{H}(\mathbb{E}_{x_b}[f(\alpha(x_b))])\\
&= l_s + \lambda_{ef}\sum_{j=1}^{c}\{(\frac{1}{B}\sum_{b=1}^{B}f_j(\alpha(x_b)))\log(\frac{1}{B}\sum_{b=1}^{B}f_j(\alpha(x_b)))\},
\end{aligned}
$$

---

[1] In fact, the class of both labeled and unlabeled data are equally distributed in reconstructed batches, but it is unrealistic to use the prior knowledge about labels of unlabeled data. Although it is fine for "debugging" the training behavior of FixMatch, when aiming at improving the performance of FixMatch, we cannot use the information about labels of unlabeled data, because it is very likely to have unbalanced classes of unlabeled data in practice. Then it makes no sense to regularize the outputs of unlabeled data in the training phase.

where $c$ is the number of classes and $\lambda_{ef}$ is a hyperparameter. We also consider the confidence entropy loss regularization which can avoid over-confident prediction,

$$
\begin{aligned}
l_s'' &= l_s - \lambda_{ce}\mathbb{E}_{x_b}[\mathcal{H}(f(\alpha(x_b)))] \\
&= l_s + \lambda_{ce}\frac{1}{B}\sum_{b=1}^{B}\{\sum_{j=1}^{c} f_j(\alpha(x_b))\log(f_j(\alpha(x_b)))\}, \\
l'' &= l_s'' + \lambda_u l_u.
\end{aligned}
\tag{6}
$$

Note that since the loss function (6) aims for avoiding over-confident predictions, it seems to be fine to regularize the unlabeled data as well. However, we cannot do that for the same reason as the loss function (5) which has been discussed in the footnote. Because $-\mathcal{H}(\cdot)$ is a convex function, we have the Jensen's inequality

$$
-\mathcal{H}(\mathbb{E}_{x_b}[f(\alpha(x_b))]) \le -\mathbb{E}_{x_b}[\mathcal{H}(f(\alpha(x_b)))].
$$

In other words, confidence entropy regularization can implicitly regularize the equal frequency of the data labels. Therefore, with the same reason, we should only apply it to the labeled data of which label distribution is under our control with augmentation.

## 4 Data Preprocessing and Augmentation

FixMatch requires a weak augmentation $\alpha(\cdot)$ and a strong augmentation $\mathcal{A}(\cdot)$. For the weak augmentation, we randomly flip an image with probability $0.5$ as [Sohn et al., 2020] and translate an image up to $12.5\%$ with probability $0.5$ [2]. For the strong augmentation, FixMatch uses either RandAugment (RA) [Cubuk et al., 2020] or CTAugment [Kurakin et al., 2020] for their experiments. However, we use RA for our experiments with the maximum magnitude 10 (same as the official experiment setup) and 2 randomly selected operations per image.

Due to the limitation of computational resources, we examine the reproducibility of [Sohn et al., 2020] on the dataset CIFAR-10 [Krizhevsky et al., 2009]. In CIFAR-10, there are 50000 training data and 10000 testing data. We take 5000 training data as the validation dataset. Then we use the remaining training dataset to make labeled and unlabeled datasets and augment both datasets into the same target number as in [Sohn et al., 2020]. After augmentation, we have $2^{13}$ labeled images and $2^{13} \times 7$ unlabeled images for the CIFAR-10 training dataset.

## 5 Experiment

In the reproducibility experiments, we re-implement FixMatch from scratch using PyTorch and reproduce the essential experiments in the original paper with the similar results. We use the hyperparameters ($\lambda_u = 1$, $\eta = 0.03$, $\beta = 0.9$, $\tau = 0.95$, $\mu = 7$, $B = 64$, $K = 2^{20}$) given by [Sohn et al., 2020] and focus on reproducing the performance on CIFAR-10 (Sec. 4.1 of [Sohn et al., 2020]) and barely supervised learning experiments (Sec. 4.4 of [Sohn et al., 2020]). Besides the early introduced hyper-parameters, we use SGD with $\beta = 0.9$ for training the model, and the learning rate is decay with $\eta\cos(\frac{7\pi k}{16K})$, where $K$ is the total time step and $k$ is the current time step. Each experiment takes around 68 hours on a single V100. And all the error rates is generated from EMA (exponential moving average) of models in the SGD training trajectory.

Then, we investigate confirmation errors of "Match"-based SSL methods to see whether there exists such error and asymmetric noise of pseudo labels in FixMatch with the official experiment setup, i.e. unbalanced batches, in [Sohn et al., 2020]. Next, we examine the existence of confirmation errors and asymmetric noise for FixMatch again in a more reliable way using re-constructed batches as introduced in Sec. 3. Furthermore, we respectively add the equal-frequency entropy regularization and confidence entropy regularization on the labeled training data in the loss function and compare with the baseline FixMatch without entropy regularization on the BC batches. Finally, we add asymmetric noise to the labeled data in the training dataset and compare the performance of baseline FixMatch and FixMatch with different entropy regularization.

---

[2] Here, [Sohn et al., 2020] didn't indicate what probability they use for the translation.

## 5.1 Reproducibility

**CIFAR-10.** We reproduced the experiments on CIFAR-10 with 40, 250, 4000 labeled data and 5000 validation samples as the official implementation of FixMatch[3]. But due to the limitation of computational resources, we didn't reproduce 5 "folds". Thus, our result based on 1 fold doesn't have the standard deviation. Our model uses the Wide ResNet-28-2 [Zagoruyko and Komodakis, 2016] with leaky ReLU activation function. Our results are shown in Table 1 which is comparable to the performance in [Sohn et al., 2020].

Table 1: Error rates for CIFAR-10 on test data. FixMatch (RA) uses RandAugment [Cubuk et al., 2020]. BC means that the experiment uses balanced-class batches as introduced in Sec. 3. We use the experiment with BC and RA as a comparison baseline results for the investigation in Sec. 5.3.

| | CIFAR-10 | | |
|---|---|---|---|
| Method | 40 labels | 250 labels | 4000 labels |
| Official FixMatch (RA) | $13.81 \pm 3.37$ | $5.07 \pm 0.65$ | $4.26 \pm 0.05$ |
| Ours (RA) | 10.04 | 5.29 | 4.36 |

**Barely supervised learning.** We also reproduce the one example per class experiment. [Sohn et al., 2020] hypothesize that the repressiveness of the chosen labeled data influences the results significantly. Since there are only one/few samples per class, this hypothesis is reasonable intuitively. Then, Sohn et al. [2020] categorized the training dataset into eight levels of "prototypicality", i.e., representative of the underlying class and then ordered the training samples by their "prototypicality". With the same hyperparameters, the model is trained with 10 provided most representative labeled data under Random Augment. The accuracy is $84.41\%$ compared with the official implementation: a median of $78\%$ accuracy and a maximum of $84\%$ accuracy.

## 5.2 Ablation studies

The ablation studies are based on FixMatch with 250 labels using CTAugment.

**Study for Confidence threshold.** We performance the ablation studies for confidence threshold. Due to the limited computation resource, we hypothesize that experiments with lower confidence threshold will achieve worse performance and explore more values around the optimal value of confidence threshold, 0.95 chosen by the authors. Thus our examined threshold value is between 0.7 to 0.98. As shown in Figure1 (c), the error rate is between $6.54\%$ and $6.19\%$ and the highest performance is under the threshold 0.98.

**Ratio of unlabeled data.** We perform FixMatch under different ratios of unlabeled data. Figure1 (d) shows the error rate which is decreasing when the ratio of unlabeled data is higher. A significant increase of the accuracy happens using a large number of unlabeled data. The results show the consistency with the finding in the original paper.

## 5.3 Investigation on confirmation errors and asymmetric noisy (pseudo) labels

In this section, we show the investigation on confirmation errors and asymmetric noise in labels and pseudo labels and whether the entropy regularization in loss functions (5) and (6) can deal with them and improve the performance of FixMatch. The training dataset contains 150 labeled data before augmentation and each BC batch in the training phase contains images with uniformly distributed classes.

**Existence of asymmetric noise and confirmation errors in pseudo labels.** We examine the existence of asymmetric noise in pseudo labels by checking the confusion matrix of the prediction of unlabeled data in different batches. Top figures of Figure 2 show the confusion matrices in the experiments without using BC batches. We find that asymmetric noise appears in a random manner, which is as our expectation as analyzed in Sec. 3. The stochastic behavior is inherited from the

---

[3]The official implementation: `https://github.com/google-research/fixmatch`. From the reproducibility and readability, the official code is not a valid submission.

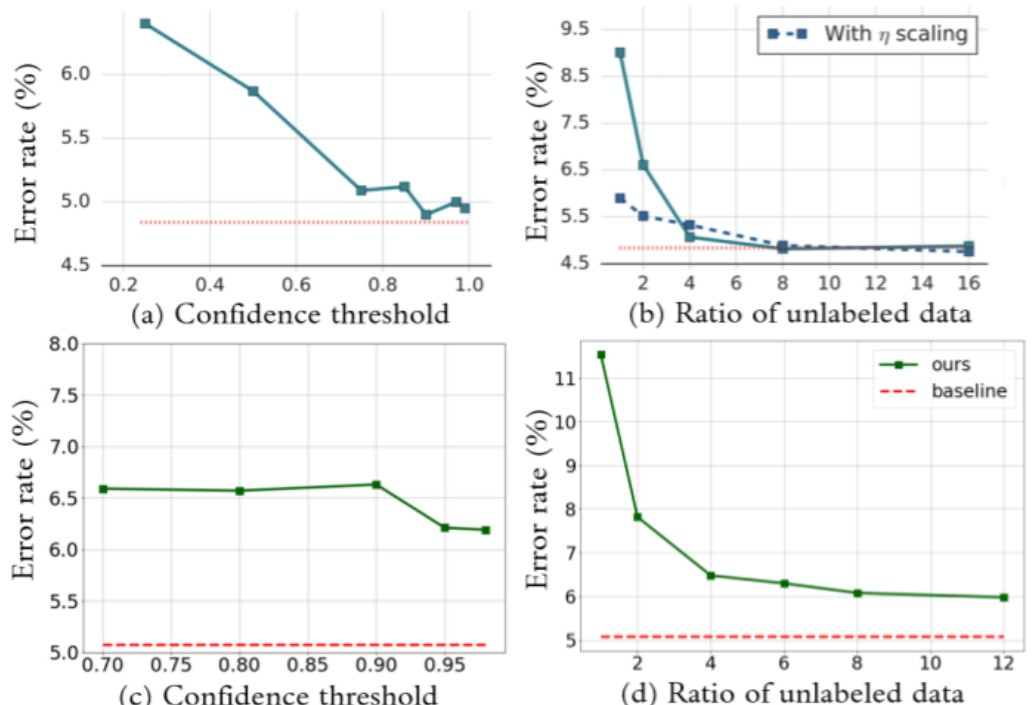

Figure 1: Plots of various ablation studies on FixMatch compared with those reported in the paper. (a) Varying the confidence threshold for pseudo-labels in the original paper. (b) Varying the ratio of unlabled data ($\mu$) in the original paper. (c)Varying the confidence threshold for pseudo-labels based on our implementation. (d) Varying the ratio of unlabeled data ($\mu$) based on our implementation.

randomness of batch composition. Next, we evaluate the asymmetric noise with BC batches, which is a more reliable way as mentioned in Sec. 3. We found that there exists consistent asymmetric noise, which leads to the confirmation errors, i.e., the model always tends to wrongly predict certain images into certain classes as shown in bottom figures of Figure 2. Moreover, the accuracy of our implementation is $93.6\%$ without BC batches and $93.8\%$ with BC batches, which shows that using BC batches has rarely influence on the model performance compared with the one without BC batches.

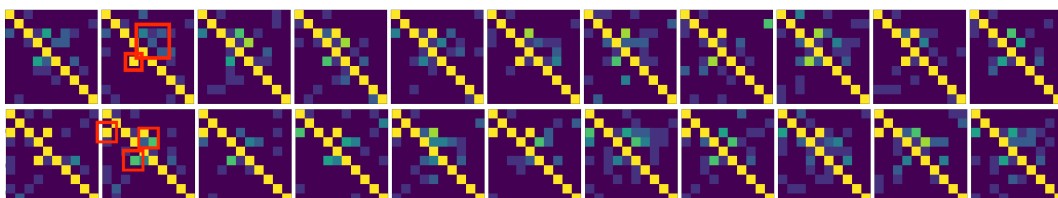

Figure 2: Confusion matrices of the confident prediction on unlabeled data with different batch structures. Confusion matrices are plotted every 100 training steps in the 1st epoch (1024 steps). The **top** matrices are from the experiments without BC, and the **bottom** matrices are the ones with BC. The red areas represent the asymmetric noise in the pseudo labels. The bottom matrices have a stable and smooth transition while the top matrices have a fluctuating transition in the red areas. The yellow color represents larger value and the darker green color represents smaller values.

**Equal-frequency and confidence entropy regularization on the labeled data.** Due to limitation of the computational resources, we didn't explicitly run grid search for the hyperparameters in the Equal-Frequency (EF) loss function (5) and Confidence-Entropy (CE) loss function (6). Instead, we found that for the baseline method the training loss is around 0.2. We then compute the equal-

frequency entropy loss for the ideal scenario, equal frequency for all classes, which is $0.1 \times \ln 0.1 \approx 2$. We decide to try the hyperparameter $\lambda_{ce}$, $\lambda_{ef} \leq 0.1$ to avoid making the entropy regularization loss dominate the loss value. Then, we do a hyper-parameter search for the loss function (5) and (6). For all experiments in this experiment, we used cosine function decay for the parameters $\lambda_{ce}$ and $\lambda_{eq}$, which starts with value $1$ and ends with value $0$ in the training phase. We find that using the loss function (6) can achieve a better accuracy performance $94.01\%$. Moreover, as an advantage, using the confidence entropy regularization can reduce the asymmetric noise as shown in the bottom confusion matrices of Figure 3. As for the equal-frequency entropy regularization, it has a better accuracy, $93.85\%$. Moreover, the equal-frequency entropy regularization can penalize the asymmetric noise, which may transform it into symmetric noise as shown in the middle confusion matrices of Figure 3. Note that there are plenty of ways to deal with symmetric noise, which is much easier to handle.

Table 2: Error rates on testing data using the loss function (5) and (6). The experiments use 150 labeled data and CTA for training. The first column is the results without BC batch and the second column is the baseline result without using EF or CE regularization.

| Entropy regularization | noBC+Null | BC+Null | BC+CE | BC+EF |
|---|---|---|---|---|
| $\lambda_{ce}/\lambda_{ef}$ | 0 | 0 | 0.1 | 0.1 |
| Error rate | 6.4 | 6.2 | **5.99** | 6.15 |

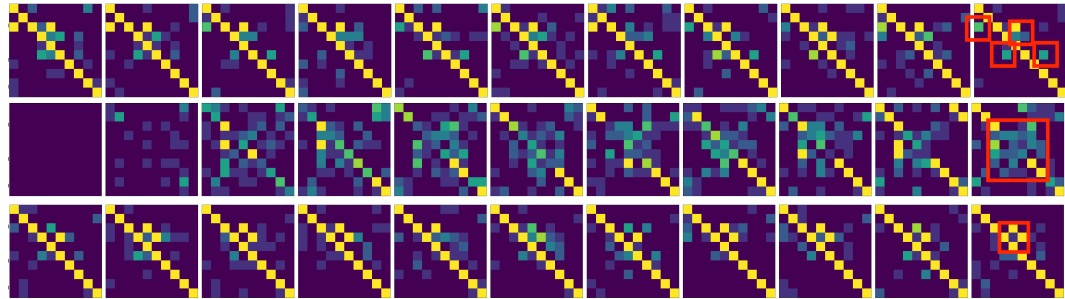

Figure 3: Confusion matrices of the confident prediction on unlabeled data with BC batches using loss functions (4) without entropy regularization at **top**, (5) with equal-frequencey entropy regularization in the **middle**, and (6) with confidence entropy regularization at **bottom**. Confusion matrices are plotted every 100 training steps in the 1st epoch (1024 steps). The red areas represent the asymmetric/symmetric noise in the pseudo labels. The yellow color represents larger value and the darker green color represents smaller values.

**Equal-frequency and confidence entropy regularization on the labeled data containing asymmetric noise.** In this experiment, we use RA data augmentation and manually add asymmetric noise to the labeled data in the training dataset to compare how FixMatch with different loss functions performs in the presence of asymmetric noise in the labeled data. We respectively select 3 images from class 0 and class 1 in the validation dataset. Then, for the labeled data in the training dataset, we keep the labels unchanged and replace 3 images in class 2 with the 3 images in class 0. Similarly we replace 3 images in class 3 with the 3 images in class 1. In this way, the only difference with the previous experiments in this section is that our final validation dataset has 4994 images and the labeled data in the training dataset contain asymmetric noise. Table 3 shows error rates on 6 runs with different random seeds. In the presence of asymmetric noise in labeled training data, all proposed methods perform better than the baseline method, in which FixMatch with BC batches decreases the average error rate from 8.6 to 7.37, and the combination of confidence-entropy regularization and BC batches further lowers the error rate to 6.98.

## 6 Challenges

It is not clear how many steps are there in each epoch. First the paper only states the total steps $K = 2^{20}$ and the composition of one batch ($B$ labeled samples and $\mu B$ unlabeled samples). And the official code indicates there are $2^{16}$ labeled images observed by the model per epoch and a total of

Table 3: Error rates of FixMatch methods in the presence of asymmetric noise in labeled training data augmented by RA: The baseline method ($\lambda = 0$); The method ($\lambda = 0$) with BC batches; the method with confidence-entropy regularization ($\lambda_{ce} = 0.1$) and BC batches; the method with equal-frequency regularization ($\lambda_{ef} = 0.1$) and BC batches.

| | $\lambda = 0$(noBC) | $\lambda = 0$(BC) | $\lambda_{ef} = 0.1$(BC) | $\lambda_{ce} = 0.1$(BC) |
|---|---|---|---|---|
| Error rates on test data | $8.6 \pm 2.81$ | $7.37 \pm 2.05$ | $7.95 \pm 2.2$ | $\mathbf{6.98 \pm 1.83}$ |

$2^{26}$ images observed which suggests that there are $2^{12}$ updates per epoch and $2^{19}$ updates in total. And this is not consistent with the total update steps $K$ stated in the paper. When performing weak augmentations to the input data, the probability for randomly translating images is not specified. And it also remains unclear the '5 different folds' mentioned in the paper, we are guessing it is a kind of cross validation while there is not too much evidence supporting this neither in the paper nor in the official code.

The paper doesn't contain sufficient details to reproduce all the experiments. Thus, it is necessary to look for details about reproducing the experiments in the official code. We have not optimized or tuned the hyperparameters, and all the hyperparameters are the same as those mentioned in the paper. Compared to the average error rates in the original paper, the reproduced results have a reasonable good performance on a larger number of labeled data (4000/250 labels) and better but also reasonable performance on fewer labeled data (40/10 labels) since the variance of error rates over 5 different folds for CIFAR-10 with 40 labels is $3.35\%$. Moreover, to compare with the results of ablation studies in the original paper, we also implement CTAugment, which supports a learnable magnitude. While we failed to confirm the result that CTAugment behaves better than RandAugment on CIFAR-10. We hypothetically guess this is because it could affect the consistency regularization because of different levels of distortions controlled by magnitude.

# 7 Conclusion

In this work, we study and reimplement FixMatch from scratch. We reproduced essential experiments, included the model performance on CIFAR 10, barely supervised learning, and ablation studies. Experimental results show that our implementation achieves similar performance as the original FixMatch results, which supports that FixMatch outperforms semi-superivesed learning benchmarks and that the author's choices with respect to those ablations were experimentally sound. We also confirmed the existence of confirmation errors in pseudo labels by checking the prediction confusion matrix of unlabeled data in different training stages. We adapted the training batch structure to be composed of equal number of images in each class, which enable us to stably and reliably check the the asymmetric noise in the training phase. Based on the reconstructed batch structure, we used the equal-frequency and confidence entropy regularization in the loss function, and theoretically show their relation. The experiments indicate that these entropy regularization can reduce the asymmetric noise in pseudo labels and improves the performance of FixMatch in the presence of training labels with asymmetric noise.

# 8 Ethical consideration

The bias in the collected dataset is a serious problem when applying machine learning methods to the real-world scenarios. For example, applying machine learning methods to making automated decision-making systems for criminal prediction, university admission or recruitment. In these cases, we may very likely collect a dataset containing certain bias due to the historical reason or selection bias in the data collection process. If a model cannot deal with such bias in the dataset, it may inherit in the model by focusing on the unrelated or wrong relations in the dataset. Consequently, the model can make biased decision which can disadvantage a certain group of people and may even diminish this group in the society.

Unfortunately, FixMatch cannot only be influenced by the noise in the label of a training dataset, but also it can make confirmation errors causing a biased model even when the dataset itself is unbiased. To deal with such issue, this work focuses on the asymmetric noise in the data labels and pseudo

labels, which can lead to severe confirmation error and the biased model. And then, we applied different methods to reduce such noise in pseudo labels and reduce its impact on the model.

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
