# OpenReview forum: "Reimplementation of FixMatch and Investigation on Noisy (Pseudo) Labels and Confirmation Errors of FixMatch"
_ML_Reproducibility_Challenge/2020 — RC2020_

### Official Review · AnonReviewer2 · 2021-02-12

**Rating:** 8
**Confidence:** 4

**Review:**

This submission reproduces the self-supervised learning results of FixMatch on CIFAR-10 and studies how the interaction between supervised and unsupervised learning objectives might lead to confirmation errors.

**Reproducing results.** The submission successfully reproduced SSL results on CIFAR-10, with error rates within the ranges provided in the original paper (c.f. Table 1). Authors reimplemented the method using Pytorch (the official code used TensorFlow), following the method description in the paper and checking the official code when something was unclear in the manuscript. I believe that reproducing FixMatch using a different deep learning framework is an important contribution to the community.

**Beyond reproducing results.** Authors devote an important part of the submission to their hypothesis that FixMatch might suffer from confirmation errors, including an exhaustive literature review. This goes beyond reproducing results and changing some hyperparameters, and can be seen as an improvement to the original method. Unfortunately, it is unclear whether the reported gains (e.g. Table 3) are statistically significant due to the lack of cross-validation or additional random seeds. The method has potential for a future workshop or conference submission if more experiments to back up the hypotheses are reported, thus I encourage authors to pursue these ideas.

Given that the submission not only reproduces some of the results of FixMatch, but also explores some of its potential limitations and proposes improvements to the method, I recommend its acceptance.

Minor comment: the first half of the paper provides error rates, but then accuracy becomes the metric of reference. While both are essentially measuring the same thing, I would encourage consistency by using the same metric throughout the entire manuscript.

**Familiar With The Original Paper:**

I have read the original paper

**Reproducibility Summary:**

Report has summary

---

### Official Review · AnonReviewer3 · 2021-03-01
**Good Summary of the original work, and reproduction of results**

**Rating:** 7
**Confidence:** 3

**Review:**

 I can confirm that the authors include a clear summary of their work. They perform all their experiments using the CIFAR-10 dataset and highlight that the original paper was missing some implementation related detailed, which was later clarified. The authors use the same hyper-parameters, as the original paper and perform some ablation studies for the confidence threshold. The authors reproduce the experiments using pytorch and do not provide any recommendations to the original authors. The paper is otherwise well written, though slightly confusing at times.

**Familiar With The Original Paper:**

I have not read the original paper

**Reproducibility Summary:**

Report has summary

---

### Official Review · AnonReviewer1 · 2021-03-08
**Review for 'FixMatch and Investigation on Noisy Labels and Confirmation Errors of FixMatch'**

**Rating:** 7
**Confidence:** 3

**Review:**

The report aims to reproduce the results of the paper 'FixMatch: Simplifying Semi-Supervised Learning with Consistency and Confidence' The report gives a summary of how the reproduction is conducted and briefly introduce the original paper. The paper also gives the details including the hyper-parameters and computational infrastructures.

The authors provides the reproducing codes with detailed documentation.

**Familiar With The Original Paper:**

I have not read the original paper

**Reproducibility Summary:**

Report has summary

---

### Decision · Program_Chairs · 2021-03-31

**Decision:**

Accept

**Comment:**

Reproducing FixMatch using a different deep learning framework is an important contribution to the community. The heat maps and figures present the results in a visual and succint way, with an important emphasis on error rates